# Dual-Specificity Phosphatase 1 (DUSP1) Has a Central Role in Redox Homeostasis and Inflammation in the Mouse Cochlea

**DOI:** 10.3390/antiox10091351

**Published:** 2021-08-25

**Authors:** Jose M. Bermúdez-Muñoz, Adelaida M. Celaya, Ángela García-Mato, Daniel Muñoz-Espín, Lourdes Rodríguez-de la Rosa, Manuel Serrano, Isabel Varela-Nieto

**Affiliations:** 1Institute for Biomedical Research “Alberto Sols”, Spanish National Research Council-Autonomous University of Madrid (CSIC-UAM), 28029 Madrid, Spain; acelaya@iib.uam.es (A.M.C.); agarciamato@iib.uam.es (Á.G.-M.); lrodriguez@iib.uam.es (L.R.-d.l.R.); 2Rare Diseases Networking Biomedical Research Centre (CIBERER), CIBER, Carlos III Institute of Health, 28029 Madrid, Spain; 3CRUK Cambridge Centre Early Detection Programme, Department of Oncology, University of Cambridge, Hutchison/MRC Research Centre, Cambridge CB2 0XZ, UK; dm742@cam.ac.uk; 4Hospital La Paz Institute for Health Research (IdiPAZ), 28029 Madrid, Spain; 5Institute for Research in Biomedicine, Barcelona Institute of Science and Technology (BIST), 08028 Barcelona, Spain; manuel.serrano@irbbarcelona.org; 6Catalan Institution for Research and Advanced Studies (ICREA), 08010 Barcelona, Spain

**Keywords:** hearing, N-acetylcysteine, glutathione, antioxidants, reactive oxygen species (ROS), mitochondria, apoptosis, inflammation, RNAseq

## Abstract

Stress-activated protein kinases (SAPK) are associated with sensorineural hearing loss (SNHL) of multiple etiologies. Their activity is tightly regulated by dual-specificity phosphatase 1 (DUSP1), whose loss of function leads to sustained SAPK activation. *Dusp1* gene knockout in mice accelerates SNHL progression and triggers inflammation, redox imbalance and hair cell (HC) death. To better understand the link between inflammation and redox imbalance, we analyzed the cochlear transcriptome in *Dusp1*^−/−^ mice. RNA sequencing analysis (GSE176114) indicated that *Dusp1*^−/−^ cochleae can be defined by a distinct profile of key cellular expression programs, including genes of the inflammatory response and glutathione (GSH) metabolism. To dissociate the two components, we treated *Dusp1*^−/−^ mice with N-acetylcysteine, and hearing was followed-up longitudinally by auditory brainstem response recordings. A combination of immunofluorescence, Western blotting, enzymatic activity, GSH levels measurements and RT-qPCR techniques were used. N-acetylcysteine treatment delayed the onset of SNHL and mitigated cochlear damage, with fewer TUNEL^+^ HC and lower numbers of spiral ganglion neurons with p-H2AX foci. N-acetylcysteine not only improved the redox balance in *Dusp1*^−/−^ mice but also inhibited cytokine production and reduced macrophage recruitment. Our data point to a critical role for DUSP1 in controlling the cross-talk between oxidative stress and inflammation.

## 1. Introduction

Progressive sensorineural hearing loss (SNHL) is a bilateral and gradual impairment of hearing as a consequence of cochlear degeneration, most commonly resulting from the death of mechanosensitive hair cells (HCs) or/and spiral ganglion neurons (SGNs), which are both irreplaceable in mammals. Genetic inheritance in combination with environmental and lifestyle factors determines the onset, severity and progression of SNHL [1], which is the most frequent sensory impairment in the elderly. The World Health Organization estimates that over 700 million people worldwide will experience disabling hearing loss by 2050 [2]. Preventive therapies to suppress the molecular mechanisms responsible for HC and SGN death are the most promising treatment options to avert irreversible damage.

Oxidative stress and mitochondrial dysfunction are common triggering factors for cochlear degeneration in all scenarios causing SNHL [3]. Accordingly, a large number of compounds have been tested for their ability to reduce reactive oxygen species (ROS) generation, promote the ROS scavenger system, or enhance cochlear antioxidant defenses Among them, N-acetylcysteine (NAC) is one of the most widely investigated, and it is administered in mice by either intraperitoneal injection or oral gavage (12.5–400 mg/kg) [4,5]. NAC oral intake is approved for human use in the range 200–1200 mg/day [4]. NAC can stimulate, indirectly, glutathione (GSH) synthesis, acting as L-cysteine precursor, and can also directly break disulfide bonds, acting as a reducing agent [6]. While the use of antioxidants to restore the redox imbalance in the cochlea has proved to be effective to some extent, in vitro and in vivo experiments revealed that the elevated levels of ROS represent just one component of a more complex and orchestrated phenomenon involving impaired mitochondrial quality control and function, loss of proteostasis, DNA damage and genomic instability, apoptosis, inflammation and altered intercellular communication [7,8]. Progress in this area is, accordingly, dependent on identifying as many molecular targets as possible in the aberrant regulatory pathways. In this context, dual-specificity phosphatase 1 (DUSP1) emerges as a central candidate.

DUSP1 is an inducible MAP kinase phosphatase (MKP), a branch of the dual specificity phosphatase family that regulates mitogen-activated protein kinase (MAPK) activation [9]. MAPKs are specific Ser/Thr kinases that operate in a three-tier kinase cascade to integrate signals from extracellular stimuli to direct physiological responses, including proliferation, differentiation, motility, inflammation and/or survival. MAPKs integrate multiple highly heterogeneous extracellular signals, including growth factors, mitogens or cytokines secreted from neighboring cells, as well as environmental stressors such as oxidative stress, UV radiation, heat or osmotic shock. JNK1/2/3 and p38 isoforms (MAPK11/12/13/14) are the major MAPKs that participate in transducing stress stimuli and, hence, are known as stress-activated protein kinases (SAPKs) [10]. SAPKs have been reported as mediators of the cochlear stress response to aminoglycoside ototoxicity, noise insult and aging both in vitro and in vivo [7,11,12]. Because the overactivation of SAPKs leads to cell death, their inhibition has been reported to promote cell survival and hearing protection after cochlear insult in humans [13] and mice [14]. The spatiotemporal activation of SAPKs leads to cellular fate-shifting outcomes, and thus, it is tightly regulated by feedback loops that are ultimately based on the dephosphorylating capacity of DUSP1. DUSP1 is induced by stress stimuli to control the magnitude and extent of SAPK activation [9,15]. It is also able, albeit with less affinity, to dephosphorylate ERK1/2 [15]. Accordingly, DUSP1 can be considered as a converging node for several signal transduction pathways, participating in multiple physiological processes [16] that include cytokine production, macrophage recruitment and inflammation resolution [17]. Mouse DUSP1 has also been reported to have a role in glucocorticoid-mediated inflammatory repression [18], limiting cytokine production [19,20] and inhibiting NF-κβ [21].

We recently showed that mice with genetic deficiency of *Dusp1* present with exacerbated inflammation and generalized HC and SGN loss along aging, thus providing a good model of premature SNHL [9]. Indeed, DUSP1 is key in the control of the progression of SNHL and in the response to noise damage mediated by increased activity of p38α [9]. Here, we aim to deepen the understanding of the role of DUSP1 in hearing loss, which we began to unveil in our previous paper [9]. The comparative RNAseq analysis of the cochleae of wild type and mutant *Dusp1* knockout mice shown in this manuscript sustained our hypothesis that oxidation secondary to DUSP1 loss of action had a role in triggering premature hearing loss. We consider that if the redox imbalance was significant, treatment with an antioxidant should prevent hearing loss. NAC was used because its antioxidant properties are well-known and its human use approved [4]. Therefore, NAC was used as a pharmacological tool to prevent redox imbalance. An in-depth knowledge of the molecular bases of SNHL with regard to DUSP1 involvement should contribute to the development of effective therapies. In this regard, the main results of the present study are the following: (i) comparative RNA sequencing (RNAseq) data analysis of wild-type (WT) and *Dusp1*^−/−^ (KO) mouse cochleae revealed the aberrant expression of genes involved in GSH metabolism, inflammation, DNA damage, apoptosis and neuronal homeostasis in *Dusp1*^−/−^ mice; (ii) antioxidant supplementation (NAC) in *Dusp1*^−/−^ mice partially recovers hearing thresholds, likely by restoring the cellular antioxidant system and limiting cochlear damage; (iii) cochleae of NAC-treated *Dusp1*^−/−^ mice show preserved mitochondrial integrity and limited ROS production, thus preventing a chronic inflammatory state; and (iv) DNA damage and apoptosis are reduced in SGNs and HCs, respectively, in NAC-treated *Dusp1*^−/−^ mice.

## 2. Materials and Methods

### 2.1. Mice

*Dusp1*^+/+^ (WT), *Dusp1*^−/−^ (KO) and NAC-treated *Dusp1*^−/−^ (KO + NAC) mice were used in the study (*n* = 25 for WT and KO, *n* = 15 for KO + NAC). Mice were generated on a mixed 129S2/SvPas:C57BL/6 genetic background and were genotyped as described [22]. No sex-linked differences were noted between genotypes in hearing; thus, male and female mice were analyzed together.

### 2.2. Antioxidant Treatment

For drug administration, we used the protocol of a previous study [23], with some modifications. Briefly, fresh NAC was prepared every week at a dose of 10 g/L in drinking water (pH ~ 7.4) and was administered for 13 weeks, beginning treatment at weaning (week 4) up to 4 months of age (week 16). Animals were maintained under controlled temperature and light–dark cycle conditions and received standard chow diet and water ad libitum. To assess the impact of antioxidant treatment on auditory function, hearing thresholds were evaluated by the auditory brainstem response (ABR) at the end of weeks 8 and 16. WT and KO mice were sacrificed, and samples were collected at both time points, whilst KO + NAC only at the second time point.

### 2.3. Hearing Evaluation

Animals were anesthetized and placed cater-cornered in a sound-attenuating chamber for recording in open field configuration 10 cm apart from the calibrated MF1 speaker. ABR testing was performed using a RZ6-A-P1 processor (Tucker-Davis Technologies, Alachua, FL, USA). Electrical auditory processing was collected in response to TDT click (broadband) and tone burst (4, 8, 16, 24 and 32 kHz) stimuli. All stimuli were presented at 31 pps rate with 10 ms acquisition time and averaged 1000 or 750 times for the click and pure tone stimuli, respectively. The response was recorded essentially as reported [24], in 5 to 10 dB steps from maximum (90 dB SPL) to minimum (15–20 dB SPL) amplitude, and analyzed using BioSigRZ software 5.6.0 (Tucker-Davis Technologies, Alachua, FL, USA). The lowest intensity (SPL level) that evoked a recognizable ABR five-peak wave pattern was established as the hearing threshold for each stimulus. ABR waves I, II, III and IV and inter-peak I-II, II-IV and I-IV latencies were analyzed at fixed 70-dB SPL click stimulation. Wave I amplitude and latency were further analyzed at all intensity levels.

### 2.4. RNA-Sequencing

Inner ear dissection was performed as described [25] and samples were frozen in RNAlater^®^ solution (Ambion, Foster City, CA, USA). Cochlear RNA was extracted using the RNeasy Plus Mini kit (Qiagen, Hilden, Germany) automated on the Qiacube (Qiagen, Hilden, Germany). For RNA sequencing, 1 μg of total RNA from cochlear RNA extracts from 20- and 32-week-old mice was used (*n* = 3 per group). The average sample RNA integrity number was 9.2–9.7 (2100 Bioanalyzer, Agilent Technologies, Palo Alto, CA, USA). The polyA+ fraction was purified and randomly fragmented, converted to double-stranded cDNA and processed through subsequent enzymatic treatments of end-repair, dA-tailing, and ligation to adapters using the Illumina TruSeq Stranded mRNA Sample Preparation Kit (Part #15031047 Rev. D; San Diego, CA, USA). This kit incorporates dUTP during the 2nd strand cDNA synthesis, meaning that only the cDNA strand generated during 1st strand synthesis is sequenced. The adapter-ligated library was completed by PCR using Illumina PE primers (8 cycles). The resulting purified cDNA library was applied to an Illumina flow cell for cluster generation and was sequenced on the Illumina HiSeq2000 platform. Single-end (50-bp) sequenced reads were analyzed with the next *presso* pipeline (http://ubio.bioinfo.cnio.es/people/ograna/nextpresso/, accessed on 1 May 2017). Sequencing quality was checked with FastQC 0.10.1 (http://www.bioinformatics.babraham.ac.uk/projects/fastqc/, accessed on 1 May 2017). Reads were aligned to the mouse genome (NCBI37/mm9) with TopHat-2.0.10 [26] using Bowtie 1.0.0 [27] and Samtools 0.1.19 [28], allowing 2 mismatches and 5 multi-hits. Transcript quantification and differential expression were calculated with Cufflinks 2.2.1 [26], using the mouse NCBI37/mm9 transcript annotations from https://ccb.jhu.edu/software/tophat/igenomes.shtml, accessed on 28 June 2021. Gene-set enrichment analysis (GSEA) [29] was conducted on Reactome, BioCarta and KEGG pathways; the RNAseq gene list was pre-ranked by statistic scores, setting ‘gene set’ as the permutation method and with 1000 permutations. Only those gene sets with significant enrichment levels (false discovery rate [FDR] *q*-value < 0.05) were considered. GSEA results were visualized using Enrichment Map [30] and AutoAnnotate [31] applications for Cytoscape 3.8.2 [32], following a described pipeline [33] and applying an FDR cutoff of 0.01 for a total of 158 enriched processes or pathways (Figure 1 and Appendix A). Heatmaps were plotted using the Heatmapper web-enabled tool [34]. RNAseq data discussed in this publication have been deposited on NCBI’s Gene Expression Omnibus and are accessible through GEO Series accession number GSE176114.

### 2.5. RT-qPCR

Inner ear dissection and RNA extraction was performed as described above. Quality determination and cDNA generation from pooled cochlear RNA extracts (3 cochlea from different animals per group) were performed as reported [35]. Quantitative amplification was performed in triplicate on a Quant Studio 7 Flex PCR System (Applied Biosystems, Foster City, CA, USA) using either commercial TaqMan probes or gene specific primers (Appendix A and [25]). Data were collected after each amplification step and analyzed with QuantStudio™ Real-Time PCR software 1.3 (Applied Biosystems). *Hprt1* gene was used as a housekeeping gene, and the n-fold differences were calculated using the 2^−ΔΔCt^ method.

### 2.6. Cochlear Morphology and Immunohistochemistry

Sample processing for histological analysis has been previously described [25]. For cochlear cytoarchitecture, inner ear paraffin cochlear cross sections (5 μm) from 3 mice per experimental group were stained with hematoxylin-eosin, analyzed with a Zeiss Axiophot microscope (Carl Zeiss, Jena, Germany) and photographed with an Olympus DP70 digital camera (Melville, NY, USA). For immunohistofluorescence, 10 μm cryostat cross sections (*n* = 3) were incubated with cytochrome c oxidase subunit I (1:100, #PA5-26688; Molecular Probes, Eugene OR, USA), p-H2A.X (1:100, #2577; Cell Signaling Technology, Danvers, MA, USA) or IBA1 (1:100, #Ab5076; Abcam, Cambridge, UK). A confocal laser-scanning (Zeiss LSM710, Carl Zeiss, Jena, Germany) microscope was used to acquire stack images. For each cochlear turn, total IBA1 intensity and mean cytochrome c oxidase subunit I intensity in the spiral ligament and spiral ganglion, respectively, were computed with Fiji software [36] from at least 3 mice of each genotype in 4 serial sections. p-H2A.X foci were counted in 5–10 neuronal nuclei per section in slides containing 4 serial sections from at least 3 mice of each genotype using the Fiji 3D objects Counter plugin in confocal stack images. To differentiate positive p-H2A.X foci from larger and less intense auto-fluorescent cytoplasmatic stress granules in neurons, high pass intensity (value = 160) and voxel size (min = 3, max = 70) thresholds were applied.

### 2.7. Hair Cell Number Quantification and TUNEL Assay

Inner ear samples were processed, dissected and stained with Myo7a (1:150, #PT-25-6790; Proteus, Ramona, CA, USA), and the TdT-mediated dUTP nick-end labeling (TUNEL) (Dead-End Fluorometric TUNEL System, Promega, Madison, WI, USA) assay was used for HC quantification and apoptosis valuation, as described [25]. Low magnification images of organ of Corti (OC) half turns were taken with a Nikon 90i epifluorescence microscope (Nikon Corp., Tokyo, Japan) and cochleograms were plotted using a custom Fiji plugin for at least 3 OC whole mounts per experimental group. HC number and TUNEL^+^ outer hair cells (OHCs) were counted in 200 μm of the basilar membrane in the apical, basal and middle regions located 15–25%, 50–60% and 70–80%, respectively, from the apex. TUNEL-positive OHCs were scored as a percentage of the total number of OHC. Representative stack images were acquired with a Zeiss LSM710 confocal laser-scanning microscope (Carl Zeiss, Jena, Germany) at the specified cochlear regions.

### 2.8. Protein Extraction and Immunoblotting

Whole cochlear pooled protein extracts were prepared from 3 mice, as described [35]. A fixed volume of extract was separated by SDS-PAGE and transferred to PVDF membranes (0.2 μm, Bio-Rad Laboratories, Hercules, CA, USA) using the Bio-Rad Trans Blot TURBO apparatus. Prior to antibody incubation (overnight 4 °C), membranes were blocked with 5% bovine serum albumin or non-fat dried milk in 0.1% Tween, 1 mM TBS. Primary antibodies used were as follows: rabbit anti-P-p38 (1:1000, #9211), anti-P-JNK (1:1000, #4668), anti-P-AKT (1:1000, #9271) and anti-P-H2AX (1:1000, #2577) (all from Cell Signaling Technology, Danvers, MA, USA), rabbit anti-P22phox (1:250, #sc-20781; Santa Cruz Biotechnology, Dallas, TX, USA), anti-HO-1 (1:1000, #AB1284; Merck, Darmstadt, Germany), anti-MnSOD (1:1000, #06-984; Millipore, Merck, Darmstadt, Germany), goat anti-NQO1 (1:1000, #ab2346; Abcam) and rabbit anti-PI3K (1:15000, in-house). Immunocomplexes were visualized with peroxidase-conjugated secondary antibodies (1 h at room temperature), and bands were detected using the Clarity™ Western ECL Substrate (Bio-Rad) on an Image Quant LAS4000 mini digital camera (GE Healthcare Bio-Sciences, Pittsburgh, PA, USA). Band densities were quantified in triplicate using Image Quant TL software 8.1 (GE Healthcare Bio-Sciences, Pittsburgh, PA, USA ).

### 2.9. Protein Carbonylation

Carbonylation of cochlear proteins was measured using the Oxyblot™ Kit (Millipore, Merck, Darmstadt, Germany). In brief, one aliquot of pooled protein extracts from each experimental group was derivatized with 2,4-dinitrophenylhydrazine (DR, derivatization reaction) and a second aliquot was treated with control solution (NC, negative control). Oxidized proteins were detected using first a primary antibody specific to the dinitrophenylhydrazone residues and then an HRP-conjugated secondary antibody. Protein extraction, electrophoresis and immunodetection were performed in triplicate as outlined above.

### 2.10. Measurement of Glucose-6-Phosphate Dehydrogenase, 6-Phosphogluconate Dehydrogenase and Glutathione Reductase Activity

Mitochondrial and cytosolic fractions from two inner ears (cochlea and vestibuli) per experimental group were isolated as described [25]. Glucose-6-phosphate dehydrogenase (G6PD) and 6-phosphogluconate dehydrogenase (PGD) activity were measured in cytosolic fractions [37], and glutathione reductase (GSR) activity was measured in both cytosolic and mitochondrial fractions using the Glutathione Reductase Assay Kit (Sigma-Aldrich, Madrid, Spain). All spectrophotometric measurements were performed at least in triplicate in a 96-well format in a VERSA max Tunable Microplate Reader (Molecular Devices, Sunnyvale, CA, USA). Protein concentrations were determined using the DC Protein Assay kit (Bio-Rad Laboratories, Hercules, CA, USA ).

### 2.11. Glutathione and Glutathione Disulfide Analysis

Measurement of GSH and glutathione disulfide (GSSG) levels was performed as described [38]. Briefly, two inner ears per experimental group were homogenized in extraction buffer (0.1% Triton X-100, 0.6% sulfosalicylic acid, 5 mM EDTA, 0.1 M potassium phosphate pH 7.5) on ice using a Dounce tissue grinder (Wheaton, Millville, NJ, USA) and centrifuged at 8000× *g* at 4 °C for 10 min. Then, 20 μL of supernatant was used for total GSH measurement or was incubated for 1 h at room temperature with 2-vinylpyridine for GSSG measurement, prior to addition of DTNB (Ellman’s reagent), glutathione reductase and β-NADPH solutions. The reaction was measured spectrophotometrically at 412 nm every 30 s for 2 min, and samples were run in triplicate in a 96-well format in a VERSA max Tunable Microplate Reader (Molecular Devices). TNB (5′-thio-2-nitrobenzoic acid) formation rates were calculated to determine total GSH and GSSG levels using linear regression from the standard curve plots. GSH concentration was obtained by subtracting GSSG from total GSH concentration. Protein concentrations were determined using the DC Protein Assay kit (Bio-Rad Laboratories, Hercules, CA, USA).

### 2.12. Statistical Analysis

Sample size was estimated to obtain a 90% statistical power with a significance level of 0.05, using data from previous experiments and calculating Cohen’s d value. Data analysis was performed with IBM SPSS 25.0 (IBM Corp., Armonk, NY, USA). Statistical significance was assessed by one-way analysis of variance (ANOVA) after the Shapiro–Wilk and Levene test for determination of normal distribution of data and equality of variances, respectively. Bonferroni or Tamhane tests were used as appropriate when differences were obtained. Results were considered statistically significant at a *p*-value < 0.05.

## 3. Results

### 3.1. Dusp1 Deficiency Affects Cellular Programs Involved in Cochlear Homeostasis

We carried out RNAseq analysis of *Dusp1*^+/+^ and *Dusp1*^−/−^ cochleae in 20- and 32-week-old mice. RNAseq datasets were used to run GSEA (Figure 1 and Appendix A). Of the 158 enriched pathways in 20-week-old *Dusp1*^−/−^ cochlea, 41 pathways were directly related to the immune system, and 17 were also enriched in 32-week-old mice. Similarly, 21 DNA-related pathways were enriched in 20-week-old *Dusp1*^−/−^ cochleae, and 12 (including DNA repair pathways) were also enriched in the 32-week-old mouse cochlear samples. Intrinsic and extrinsic apoptotic pathways were also enhanced at 20 but not at 32 weeks. Moreover, 18 pathways related to neural homeostasis were negatively enriched in *Dusp1*^−/−^ cochleae at 20 weeks, as compared with *Dusp1*^+/+^ cochleae, and eight of them (including neuronal system and glutamatergic synapse pathways) remained negatively enriched at 32 weeks. We also observed gene enrichment for the oxidative phosphorylation process in 20-week-old *Dusp1*^−/−^ cochleae, which pointed to an increase in mitochondrial ROS production by a highly active electron transport chain. By contrast, the mitochondrial biogenesis program driven by the peroxisome proliferator-activated receptor γ coactivator 1α (PGC1A) pathway was negatively enriched, suggesting the reduced generation of new mitochondrial content and impaired organelle dynamics and function. Of note, these two phenomena correlated with the enhancement of the antioxidant GSH metabolism pathway in 20-week-old *Dusp1*^−/−^ mice, which was maintained at 32 weeks of age (Figure 2A), possibly to compensate/overcome mitochondrial dysfunction and oxidative damage. Overall, these results indicate that deficiency in DUSP1 activity in mice impacts cochlear redox homeostasis, mitochondrial quality control, apoptosis and inflammation.

### 3.2. N-Acetylcysteine Treatment of Dusp1^−/−^ Mice Has a Positive Impact on Cochlear Glutathione Metabolism

We hypothesized that the gene upregulation of the GSH metabolism pathway in *Dusp1*^−/−^ cochleae might be a consequence of increased ROS production by over-functioning mitochondria that likely drives early and progressive hearing loss in *Dusp1*^−/−^ mice. Accordingly, enhancing GSH synthesis with NAC might be a strategy to counteract redox imbalance (Figure 2B). To test this, we supplied NAC (10 g/L in drinking water) to 3-week-old *Dusp1*^−/−^ mice for 13 weeks and then evaluated hearing loss progression by ABR in the three experimental groups—WT, KO and KO + NAC—at 8 and 16 weeks of age. Samples were collected at both time points for untreated groups and only at the end time point for NAC-treated *Dusp1*^−/−^ mice (Figure 2C). We found that GSH levels were higher and GSSG levels were lower in 8-week-old *Dusp1*^−/−^ inner ears than in equivalent *Dusp1*^+/+^ inner ears, resulting in a higher GSH:GSSG ratio (Figure 2D). These differences were not sustained in 16-week-old *Dusp1*^−/−^ mice, which showed lower levels of reduced and oxidized forms than *Dusp1*^+/+^ mice and similar GSH:GSSG ratio levels (Figure 2D). At this time point, inner ear GSH and GSSG levels were higher in NAC-treated *Dusp1*^−/−^ mice than in untreated *Dusp1*^−/−^ mice but were not significantly different from those in *Dusp1*^+/+^ mice. Inner ear GSR activity in both cytosolic and mitochondrial extracts was lower in 16-week-old WT and KO + NAC groups than in equivalent KO groups (Figure 2E). GSR activity in both compartments increased from 8 to 16 weeks of age in the untreated KO group (Figure 2E).

Finally, RT-qPCR analysis of genes related to cochlear GSH synthesis revealed that expression of the antioxidant enzyme glutathione peroxidase 1 (*Gpx1*) was higher in the untreated KO group than in the WT and KO + NAC groups (Figure 2F). An eventual higher GPX1 activity would require, from the higher GSR activity observed in *Dusp1*^−/−^ mice, maintaining the GSH pool and catalyzing the hydrogen peroxide reduction. Of note, NAC treatment specifically reduced the cochlear expression of the GSH recycling enzyme, gamma glutamyl transferase 1 (*Ggt1*), as compared with untreated groups (Figure 2F).

### 3.3. N-Acetylcysteine Treatment Delays Hearing Loss Onset and Reduces Hair Cell Death in Dusp1^−/−^ Mice

We next performed hearing tests in the three experimental groups at 8 and 16 weeks of age. ABR analysis showed that NAC administration protected against hearing loss in *Dusp1*^−/−^ mice, with intermediate hearing thresholds between those of untreated *Dusp1*^−/−^ mice and *Dusp1*^+/+^ mice (Figure 3A). Specifically, NAC-treated *Dusp1*^−/−^ mice and *Dusp1*^+/+^ mice displayed lower hearing thresholds for the click and tone burst stimuli than *Dusp1*^−/−^ mice at 8 and 16 weeks of age, although with evident age-related differences illustrated by the characteristic progressive high frequency hearing loss pattern in all mouse groups. NAC-mediated protection at the end-time point was evident for the frequencies located in the middle regions of the basilar membrane (8, 16 and 24 kHz) and when the cochleae were multifrequency stimulated (click stimulus). Threshold shifts between both time points (Appendix A) suggested that the antioxidant power provided by NAC was particularly effective in the first weeks of treatment, as evidenced by the faster sound conduction in 8-week-old NAC-treated *Dusp1*^−/−^, but not later, as revealed by ABR latency analysis (Appendix A).

In accordance with the hearing evaluation findings, histological analysis of cochlear sections revealed better preservation of the OC and SGNs in the basal and middle turns of 16-week-old NAC-treated *Dusp1*^−/−^ mice than of untreated *Dusp1*^−/−^ mice (Figure 3B). No anatomical alterations were observed in cochleae of *Dusp1*^+/+^ mice, whereas untreated *Dusp1*^−/−^ mice displayed a flat OC and neuronal fiber degeneration in the basal turn. NAC-treated *Dusp1*^−/−^ mice exhibited basal turn OHC loss with no alterations in the SG. Untreated-*Dusp1*^−/−^ mice showed patchy OHC loss in the middle turn that was not observed in the other two groups, as expected from the ABR analysis. Next, we performed TUNEL assays and HC Myo7a immunostaining in OC whole mounts (Figure 3C). HC counts confirmed the alterations observed by hematoxylin-eosin staining, with significant inner (IHC) and outer (OHC) loss in the basal turn cochlea of *Dusp1*^−/−^ mice as compared with NAC-treated *Dusp1*^−/−^ and *Dusp1*^+/+^ mice (Figure 3D). Accordingly, the percentage of apoptotic OHCs in the basal turn was higher in untreated *Dusp1*^−/−^ mice than in the other two groups (Figure 3E).

### 3.4. N-Acetylcysteine Administration Modulates Redox Systems and Limits Reactive Oxygen Species Production in Dusp1^−/−^ Cochleae

To better understand how NAC treatment modulates the cochlear antioxidant defense and ROS production systems in *Dusp1*^−/−^ mice, we studied the expression levels of key redox genes. Gene expression analysis showed that *Nrf2*, *Nqo1*, *Sod2* and *Prdx6* levels were higher in 16-week-old *Dusp1*^+/+^ mice than in younger mice (Figure 4A); however, the overall expression profile indicated that NAC administration did not stimulate antioxidant gene expression, which was similar between untreated and NAC-treated *Dusp1*^−/−^ mice. In fact, the expression of several genes (*Keap1*, *Sirt1*, *Sesn2* and *G6pd*) was lower in cochleae of NAC-treated *Dusp1*^−/−^ mice than in equivalent untreated mice when compared with *Dusp1*^+/+^ mice, whereas *Ho1* expression was lower in NAC-treated cochleae than in the other two groups. By contrast, *P22phox,* a component of the O_2_^·−^ generating NADPH oxidase (NOX) complex, increased in 16-week-old *Dusp1*^−/−^ mice, and this was not evident in equivalent NAC-treated *Dusp1*^−/−^ mice, which had levels similar to those of *Dusp1*^+/+^ mice. The same expression pattern was observed for *Ucp2*, an important mitochondrial carrier that controls mitochondria-derived ROS generation in the respiratory chain and whose expression is stimulated upon oxidative stress. The cochlear expression of *Ucp1* was preserved by NAC treatment in *Dusp1*^−/−^ mice as compared with untreated *Dusp1*^−/−^ and *Dusp1*^+/+^ mice, as both latter groups showed a marked reduction in expression when compared with levels in 8-week-old mice. UCP1 generates heat by leaking protons to the mitochondrial matrix and increasing mitochondrial permeability. Finally, the cochlear expression of the nitric oxide isoforms *iNos* and *nNos* was higher in 16-week-old *Dusp1*^+/+^ mice than in 8-week-old counterparts and was unaffected (*iNos*) or lower (*nNos*) in equivalent *Dusp1*^−/−^ with or without NAC supplementation (Figure 4B).

Antioxidant defense, particularly the GSH system, is highly dependent on the reductive power of NADPH, which is produced mainly in the cytosol by sequential reactions catalyzed by G6PD and PGD. Enzymatic measurements revealed higher activity of both enzymes in 8- and 16-week-old untreated and NAC-treated *Dusp1*^−/−^ mice than in *Dusp1*^+/+^ mice (Figure 4C). We also assessed the extent of oxidative damage by analyzing protein carbonylation, finding that NAC administration failed to prevent the formation of these protein modifications in 16-week-old *Dusp1*^−/−^ mice (Figure 4D). Analysis of HO1 and SOD2 protein levels revealed a different pattern of expression to that expected from the mRNA expression data, indicating the likely post-transcriptional regulation of these proteins. Of note, P22PHOX protein levels mirrored the mRNA expression data (Figure 4E), supporting the idea that NAC effectively suppresses NOX-derived O_2_^·−^ generation. Finally, to evaluate pro-apoptotic and survival signaling we studied SAPK and AKT phosphorylated levels, respectively. We observed clear activation of cochlear JNK in NAC-treated *Dusp1*^−/−^ mice compared with the other two groups, whereas AKT was slightly activated in untreated *Dusp1*^−/−^ mice as compared with *Dusp1*^+/+^ mice (Figure 4E).

### 3.5. N-Acetylcysteine Administration Protects Mitochondria and Restrains DNA Damage in Spiral Ganglion Neurons

Increased ROS generation and inefficient oxidative phosphorylation are closely linked to impaired mitochondrial function and quality control. Immunolabeling of SGN for mitochondrial complex IV (cytochrome C oxidase)—used as a marker of mitochondrial integrity—revealed evident differences between groups, with significantly stronger neuron immunostaining in NAC-treated *Dusp1*^−/−^ mice, both in the basal and middle cochlear turns, than in untreated *Dusp1*^−/−^ mice (Figure 5A,B); no differences were observed between *Dusp1*^+/+^ and untreated *Dusp1*^−/−^ mice.

To further study energy metabolism and the generation of new mitochondrial mass, we measured the expression levels of the PGC-1 family members (*Pgc1α, Pgc1β*), in addition to downstream genes and canonical mitochondrial genes (Figure 5C). Expression data indicated that the mitochondrial biogenesis program was stimulated in 16-week-old *Dusp1*^+/+^ mice, as evidenced by the higher expression levels of regulatory (*Pgc1α*, *Pgc1β*, *Pparα*, *Tfb2m* and *Alas1*) and electron transport chain component (*Sdha*, *Uqcrc2*, *Mtco1* and *Mtap6*) genes and citrate synthase (*Cs*) when compared with younger mice. This induction of gene expression was not evident in NAC-treated or untreated *Dusp1*^−/−^ mice, with the expression of most genes significantly reduced at this age in both groups (Figure 5C). This accords with the downregulation of the PGC1A pathway in *Dusp1*^−/−^ mice revealed by RNAseq (Figure 1).

Redox dysregulation may lead to DNA damage, which was highlighted in the GSEA. To evaluate this, cochlear sections were stained for the DNA damage marker phosphorylated histone 2A.X (p-H2A.X) (Figure 5D), and foci number were quantified in SGN nuclei (Figure 5E). The percentage of SGNs with >5 p-H2A.X foci per nucleus was higher in the middle turn cochlea from untreated *Dusp1*^−/−^ mice than from the other two groups. Of note, p-H2A.X levels in whole cochlear protein extracts were significantly lower in NAC-treated *Dusp1*^−/−^ mice than in untreated *Dusp1*^−/−^ mice (Figure 5F).

### 3.6. N-Acetylcysteine Administration Normalizes the Inflammatory Response and Macrophage Recruitment in Dusp1^−/−^ Cochlea

Chronic inflammation is a common feature of aberrant ROS production, which stimulates the inflammatory response in a positive feedback loop to ultimately disrupt cell and tissue homeostasis. In this context, DUSP1 is a key regulator of cell fate in response to oxidative and inflammatory stimuli. To further understand how loss of *Dusp1* impacts the regulation of these two pathways, we studied the production of inflammatory mediators in the 3 groups of mice (Figure 6A). Gene expression analysis of cochleae revealed a marked induction of proinflammatory cytokine gene expression of *Tnfα*, *Il1β*, *Il6* and *Il18* in *Dusp1*^−/−^ mice from 8 to 16 weeks. Even at the earlier age tested, cytokine expression levels were higher in *Dusp1*^−/−^ mice than in *Dusp1*^+/+^ mice with the exception of *Il18*. *Il1β* expression also increased from 8 to 16 weeks in *Dusp1*^+/+^ mice, albeit to a much lesser extent. Of note, cytokine expression levels were similar between NAC-treated *Dusp1*^−/−^ mice and *Dusp1*^+/+^ mice, indicating that NAC blunts the inflammatory response. A similar NAC response was observed for the expression levels of *Tgfβ1*, *Mpo*, *Foxp3* and *Kim1*. Overall, these results suggest that the induction of inflammatory gene expression observed in *Dusp1*^−/−^ cochleae was secondary to the oxidative imbalance, as it could be repressed, albeit incompletely, with NAC. To further confirm this, we tested for the presence of macrophages in the spiral ligament of cochlear sections by IBA1 immunolabeling (Figure 6B). Results showed stronger IBA1 staining in *Dusp1*^−/−^ basal and middle cochlear turns than in equivalent sections from *Dusp1*^+/+^ mice. NAC administration effectively limited macrophage infiltration in the middle turn but not in the basal turn (Figure 6C), which correlates well with our data on cochlear damage and hearing thresholds.

## 4. Discussion

DUSP1 appears to be a converging node for stress-related inflammation and oxidative stress pathways in the cochlea. DUSP1 contributes to redox homeostasis, inflammatory response and, consequently, to hearing preservation. Our findings indicate that administration of the antioxidant NAC to *Dusp1*-deficient mice mitigates, at least in part, the adverse effects caused by DUSP1 loss-of-function and delays the onset of hearing loss. It is likely, however, that DUSP1 has broader functions, and thus NAC administration fails to fully restore hearing in *Dusp1*-deficient mice.

Cochlear stress triggers the activation of pro-apoptotic and survival signaling pathways through MAPK cascades [9,39], and the magnitude of cross-talk between these pathways likely tips the balance in one direction or the other [40]. A uniqueness of DUSP1 is its ability to control multiple pathways and to orchestrate cellular responses to multiple stimuli [15]. Our high-throughput cochlear transcriptome profiling provides valuable data on the contribution of DUSP1 to the molecular mechanisms underlying hearing loss. As expected, pathways related to immune system were among the most affected by *Dusp1* deletion in mice, both at 20 and 32 weeks of age. Pro-inflammatory cytokines activate SAPK signaling pathways during the acute phase of the inflammatory response, which can ultimately trigger apoptosis. In this line, tumor necrosis factor-*α* (TNF*α*) is one of the most investigated factors involved in HC death pathways in vivo and in vitro—either by the intrinsic mitochondrial cascade or through the binding of ligands to cell surface death receptors (extrinsic pathway) [41]. Concomitant with immune cell recruitment, the expression of TNF*α* increases rapidly after cochlear damage, activating nuclear factor kappa-light-chain-enhancer of activated B cells (NF-κβ) and stimulating the production of other cytokines, chemokines and adhesion molecules [42]. TNF*α* and NF-κβ signaling, cytokine, chemokine, cell adhesion, phagocytosis and pathways related to intrinsic and extrinsic apoptotic cascades were prominent in the enriched pathways in cochleae from 20-week-old *Dusp1*^−/−^ mice. We speculate that *Dusp1* deletion renders the cochlea unable to resolve the pro-inflammatory response, contributing to HC and SGN apoptosis [9,17].

The importance of DUSP1 for neuronal homeostasis was also evident in our study, as illustrated by the downregulation of neuronal system pathways in *Dusp1*^−/−^ cochleae. Neuroprotective roles have previously been ascribed to DUSP1, and reduced levels have been related to neurological disorders [43]. Our data show that several pathways related to neurodegenerative diseases (including Huntington’s, Alzheimer’s and Parkinson’s disease) were enriched in the absence of DUSP1. Pathways related to DNA processes were also conspicuous in *Dusp1*^−/−^ cochlea. Among them, DNA damage pathways including DNA repair, ATM and ATR-BRCA pathways were upregulated. Because of its capacity to inhibit apoptotic signals, DUSP1 has been widely studied in vivo and in vitro as a proto-oncogene and is associated with tumor progression in several types of cancer [44], which agrees with the changes in cell cycle and cancer-related pathways in our dataset.

Mitochondrial function and maintenance of energetic integrity are essential for cellular homeostasis. Based on our data, mitochondrial quality control and biogenesis regulation in the cochlea are likely to be impaired in the absence of DUSP1 [45], as revealed by the downregulation of PGC1α and the enhanced oxidative phosphorylation pathways in 20-week-old *Dusp1*^−/−^ cochlea. Mitochondria are extremely dynamic and environmentally sensitive organelles that can adapt rapidly to adverse conditions to preserve energetic supply [46]. Even mild mitochondrial stress has been proposed to protect the cell through increased ROS signaling as an adaptative response to reduce vulnerability to additional metabolic anomalies [47]. Indeed, mitochondria produce up to the 90% of all intracellular ROS, which are associated with cellular respiration, making them one of the main targets of oxidative damage. Loss of DUSP1 function might render cochlear cells incapable of adapting to a pro-oxidant environment, leading to overwhelmed mitochondria and damage [48]. This is consistent with the sustained gene enrichment for GSH metabolism observed in 20- and 32-week-old *Dusp1*^−/−^ mice, likely induced to restore redox balance. In this sense, metabolism and inflammation are both closely linked to ROS signaling [49], which might account for the exacerbated inflammatory response observed in *Dusp1*^−/−^ mice. These findings suggest that increased ROS production and mitochondrial dysfunction trigger the inflammation and apoptotic pathways activated in *Dusp1*^−/−^ cochlea [9,39,50]. This is supported by our finding that the administration of a GSH prodrug restores redox balance and lessens the impact of DUSP1 loss-of-function for cochlear homeostasis.

In anticipation of the oxidative imbalance in *Dusp1*^−/−^ mice, we administered NAC to pups after weaning. Measurement of GSH levels indicated that the usage of GSH is finely tuned in NAC-treated cochleae, with higher total GSH levels and lower GSR turnover rate than in untreated cochleae, suggesting a reduced need for replacing GSSG to increase the GSH pool. Accordingly, *Ggt1* mRNA levels were reduced, which might indicate that NAC provides a more efficient source of cysteine than the ATP-dependent process associated with GSH extracellular export and recycling. Indeed, NAC administration has been reported before to prevent progressive hearing loss in *Ggt1* deficient mice, presumably by providing cysteine [51].

In line with the above, we found that the expression of antioxidant defense enzymes was not induced in NAC-treated cochlea samples, and no differences were observed in protein carbonylation. It is likely that the correct combination of antioxidants with different mechanisms of action, rather than NAC alone, would enhance the cochlear antioxidant defense impacting simultaneously at different levels [5]. By contrast, WT cochleae showed an age-related and coordinated induction of antioxidant genes, possibly driven by NRF2 [52], which prevented protein oxidative modifications. However, our results point to a limitation in ROS production at this age that correlated with a better preservation of mitochondrial function and energetic supply in NAC-treated cochlea. Conversely, NADPH production by G6PD and PGD was not reduced in NAC-treated cochlea, which is particularly intriguing as NOX and GSR activities depend highly on this cofactor [53]. In this regard, RNAseq data suggest that the distinct regulation of G6PD and PGD in wild type and knockout mice may be a consequence of the metabolic alterations in the pentose phosphate cycle associated with *Dusp1* deficiency. Furthermore, *Dusp1*^−/−^ cochlea showed increased transcript and protein levels of P22PHOX [25], suggesting that NADPH production could be increased to maintain the levels due to increased spend. In contrast, we speculate that neither O_2_^−^ generation (by NOX) nor GSH regeneration is required at higher rates in NAC-treated cochlea. Therefore, NADPH is likely used by other NADPH-dependent enzymes that participate in reductive anabolic reactions.

The mitochondrial biogenesis program was not activated in NAC-treated *Dusp1*^−/−^ mice in contrast to what we observed in WT cochleae. Nevertheless, mitochondria integrity was preserved in NAC-treated SGNs. On this basis, we propose that early NAC administration generates a balanced redox environment in the mitochondria that prevents the oxidative damage in the organelle [54] that would, otherwise, have a detrimental outcome, as observed in untreated *Dusp1*^−/−^ mice.

Active SAPKs promote cytokine production and leukocyte migration to injury sites. Therefore, tight control of the activation of these kinases is key to preventing an inflammatory response and the propagation of damage [17]. In this context, DUSP1 is induced in response to inflammatory stimuli to mediate shutdown of the acute inflammation phase [55]. We found that inflammation was present at early stages in *Dusp1*^−/−^ cochlea, later becoming chronic with production of inflammatory mediators that contributed to severity and propagation of damage accompanied by increased macrophage recruitment [56,57]. In this regard, expression of the master regulator of the development and function of regulatory T-cells, *Foxp3*, was induced in *Dusp1*^−/−^ cochlea at 16 weeks of age. When stimulated, regulatory T-cells aid in resolving established inflammation by suppressing immune responses of other cells and inducing them to express immunosuppressive cytokines such as IL-10 [58]. Despite the higher *Il10* expression in 16-week-old *Dusp1*^−/−^ mice, the increase was not significant and might indicate deficient activation of the inflammation resolution phase. Specifically, IL-10 is known to induce the robust expression of *Dusp1* that sustains mRNA levels in a feedback loop and correlates with inhibition of p38 signaling and with IL-6 and IL-12 production in vitro [59].

Inflammation was reduced in NAC-treated mice, although it was not completely neutralized, as illustrated by increased *Tnfα*, *Il6* and *Il18* expression levels. It is possible that NAC contributes to preventing early ROS-driven pro-inflammatory cytokine production and the later chronic inflammatory state [60,61], as observed in NAC-treated 16-week-old *Dusp1*^−/−^. Nevertheless, the cellular antioxidant system is complex, and while glutathione is the most powerful cellular antioxidant, it represents just one of the several lines of defense against ROS. In this context, the impact of NAC administration is limited and might explain the higher JNK activation levels [62] observed in NAC-treated 16-week-old *Dusp1*^−/−^ mice, which may have occurred earlier in chronically inflamed untreated *Dusp1*^−/−^ cochlea. We would suggest that HC loss may spread quickly in NAC-treated mice, emulating the situation in untreated mice and supporting the idea that hearing loss onset is lessened but not prevented by NAC. Indeed, although much lower than in untreated mice, the gene expression of the damage biomarker kidney injury molecule 1 (KIM-1) [9] was higher in NAC-treated mice than in control mice, which also points in this direction. Cochlear levels of KIM-1 have been reported to increase following NOX activation and ROS generation, and ROS scavengers and NOX inhibitors attenuated this increase [63]. Notably, *Kim1* induction upon ROS generation by NOX was also observed in the *Dusp1*^−/−^ cochlea as shown by increased P22PHOX mRNA and protein levels, which were inhibited by treatment with NAC.

Independent of ROS production, aberrant regulation of inflammation due to DUSP1 loss-of-function contributes to damage extension and hearing loss. Indeed, corticoid treatment was previously found to reduce TNFα levels in the spiral ligament and activate survival pathways in HCs, inhibiting JNK signaling [64]. In this context, modulation of both mechanisms simultaneously could be a more robust strategy to ameliorate the metabolic disturbances in *Dusp1*^−/−^ cochlea. Indeed, there is evidence showing that drug combination approaches attenuate human hearing loss [65]. In fact, protection from oxidative stress and limited HC death were reported in patients with sudden SNHL by combined NAC-dexamethasone treatment, but both compounds were ineffective when administered alone [66].

Overall, our study provides evidence to consider pharmacological modulation of DUSP1 activity as a potential treatment for hearing loss, as an alternative to current SAPK inhibitors [13,14,67,68]. Nevertheless, given its unique physiological activity, DUSP1 induction and activity regulation is finely tuned, and so treatments would be highly dependent on the correct timing and kinetics of both DUSP1 and the proteins involved in the signaling pathways. It is clear that several issues need to be addressed and numerous challenges overcome to achieve the successful modulation of DUSP1 activity [69].

## 5. Conclusions

We provide new insight into the role of DUSP1 in the cochlea using a combination of transcriptomics and antioxidant treatment of *Dusp1*-deficient mice. We present here, for the first time, transcriptome sequencing data from cochleae suffering unrestrained cellular stress due to the absence of its natural regulator DUSP1. Our data improve knowledge on the basic molecular mechanisms that orchestrate the regulation of oxidative stress, inflammation and mitochondrial quality control in the ageing cochlea. The molecular mechanisms underlying SNHL have multiple common aspects; thus, this work provides information beyond ARHL. Finally, our findings point to DUSP1 as a potentially druggable target to impair HC and SGN death in response to stress.

## Figures and Tables

**Figure 1 antioxidants-10-01351-f001:**
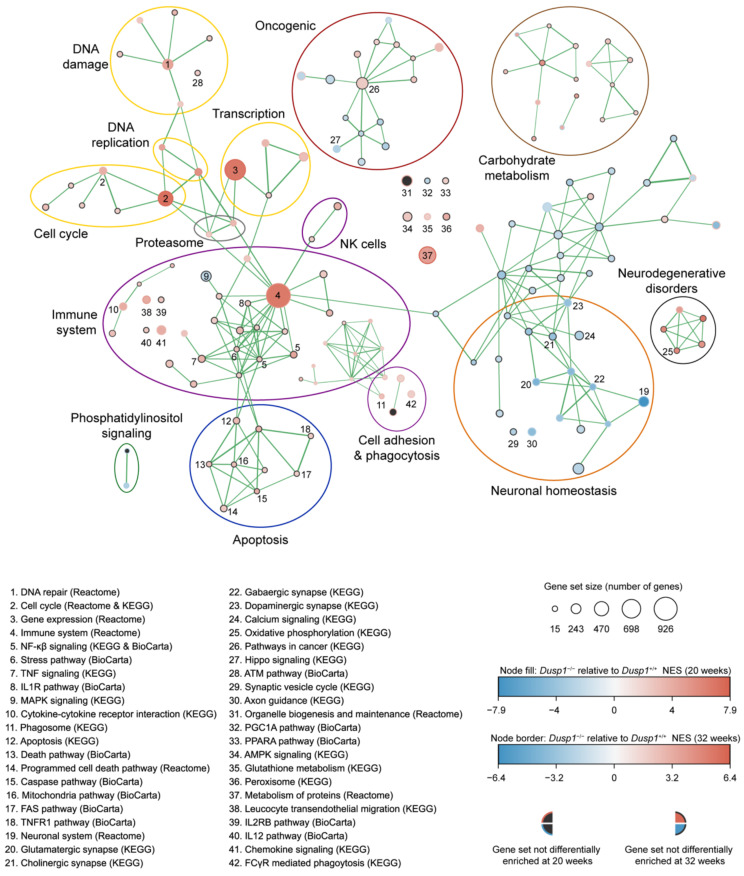
Gene-set enrichment analysis of RNAseq data sets from 20- and 32-week-old *Dusp1*^+/+^ and *Dusp1*^−/−^ cochlea. Enrichment analysis visualization for identification of altered gene sets (nodes). Gene sets derive from Reactome, KEGG and BioCarta databases. Node size denotes gene set number. Differentially enriched gene sets in 20-week-old *Dusp1*^−/−^ mice are designated by node fill while node border designates 32-week-old data. Normalized enrichment value (NES) for each time point is illustrated in a color scale from blue (negative enrichment) to red (positive enrichment). No representation of the gene set at either time point is shown by a grey node fill or grey border, respectively. Connection between nodes is represented by lines. Line width symbolizes number of genes shared by both nodes. Most interesting nodes are numbered and listed (1–42). Related nodes are grouped in modules and labeled accordingly. Related modules share the same color.

**Figure 2 antioxidants-10-01351-f002:**
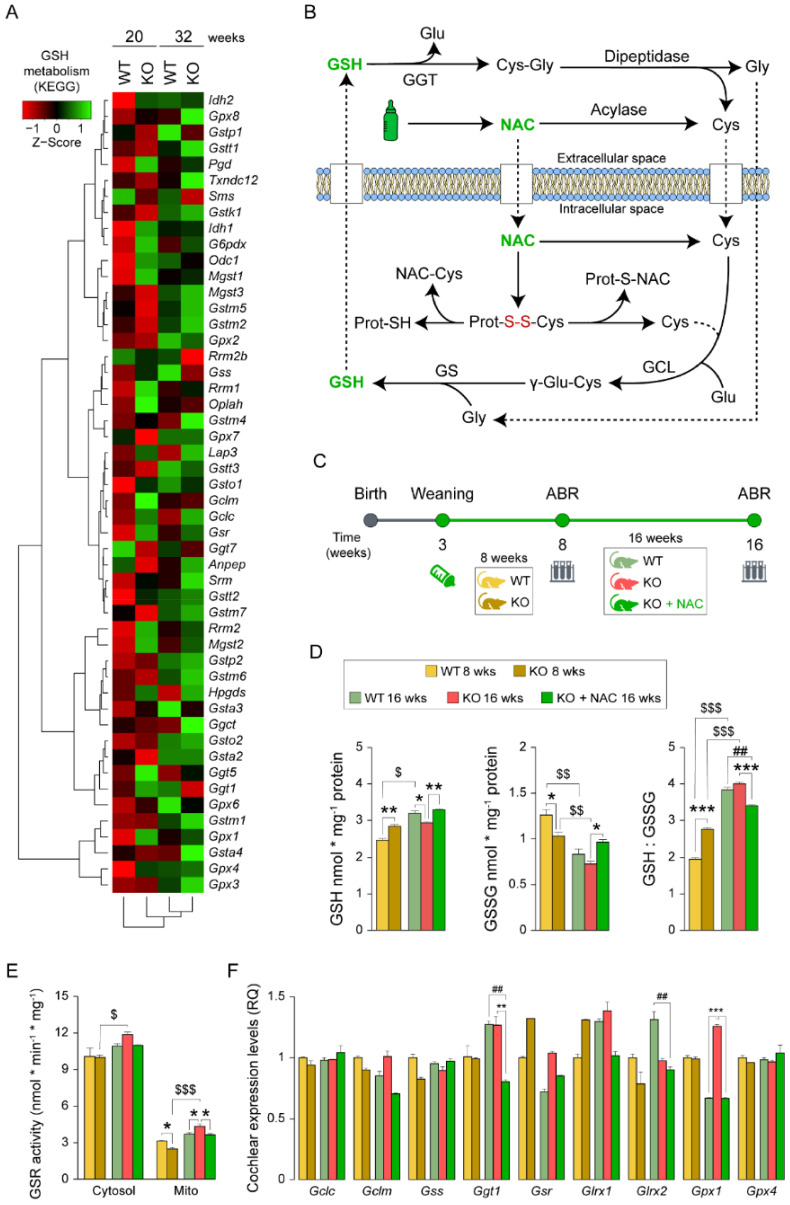
Impact of N-acetylcysteine administration for glutathione metabolism. (**A**) Glutathione (GSH) metabolism (KEGG database) heatmap generated from differential expression data derived from RNAseq. (**B**) Scheme illustrating GSH biosynthesis, extracellular recycling and N-acetylcysteine (NAC) antioxidant power. GSH and NAC appear in green; glutathione synthesis and recycling enzymes appear in black capital letters: GCL (glutamate-cysteine ligase), GS (glutathione synthetase), GGT (gamma glutamyl transferase); glutathione precursors: Cys (cysteine), Glu (glutamate) and Gly (glycine); oxidized protein disulfide bonds in red (S-S). (**C**) Experimental design illustrating the start and end-time points of NAC administration (weaning-16 weeks), hearing evaluation (auditory brainstem response [ABR]) testing time points (8 and 16 weeks), experimental groups and sampling time points. (**D**) GSH, glutathione disulfide (GSSG) levels and GSH:GSSG ratio from pooled samples of 2 inner ears of 8- and 16-week-old wild-type (WT), *Dusp1* knock-out (KO) and NAC-treated *Dusp1* KO mice. All measurements were performed in triplicate. GSH and GSSG values are presented relative to milligram of protein. (**E**) GSR (glutathione reductase) activity in cytosolic and mitochondrial fractions from pooled samples of 2 inner ears per condition. Measurements were performed in triplicate, and values are presented relative to milligram of protein. (**F**) RT-qPCR gene expression levels of GSH-related enzymes from whole cochlea pooled samples from 3 mice per condition. Expression levels were calculated as 2^−ΔΔCt^ (RQ) from triplicate measurements using *Hprt1* as a reference gene and normalized to levels in 8-week-old WT mice. All data are presented as mean ± SEM. Statistical significance between genotypes and time points was analyzed by one-way ANOVA: * vs. KO, # vs. WT, $ 8-week-old mice vs. 16-week-old mice (*, $ *p* < 0.05; **, ##, $$ *p* < 0.01; ***, $$$ *p* < 0.001).

**Figure 3 antioxidants-10-01351-f003:**
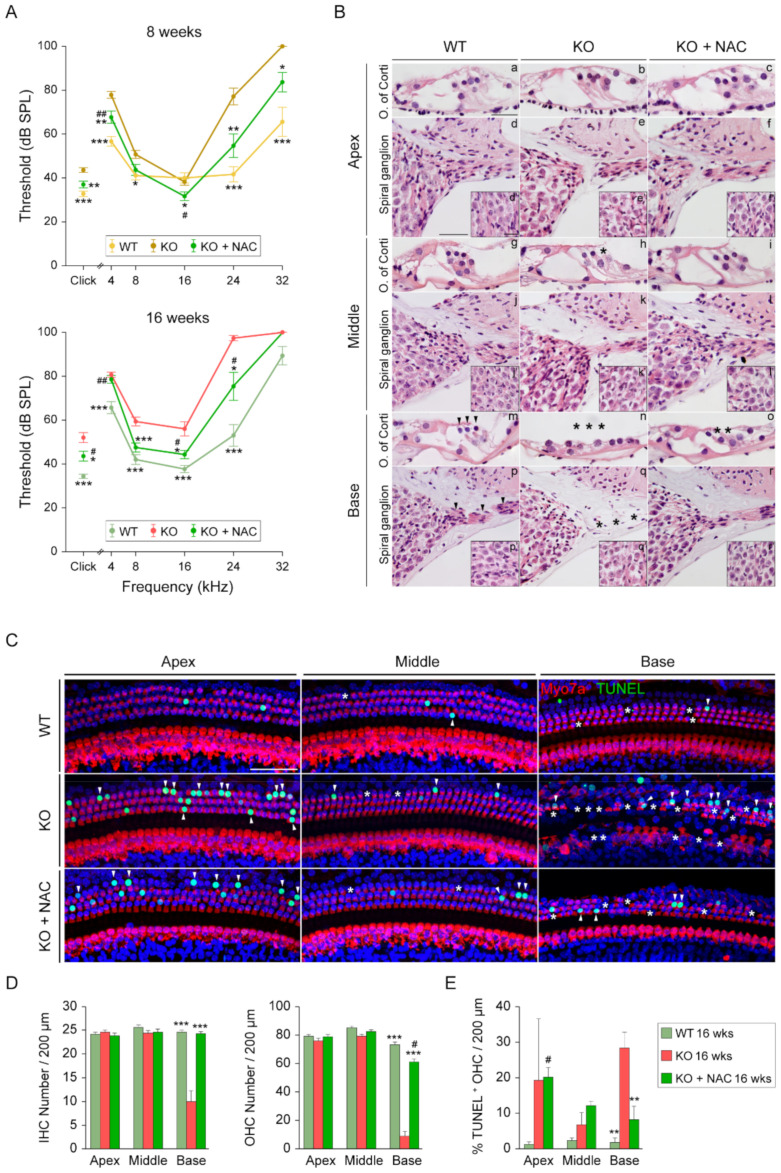
Comparative hearing evaluation, cochlear cytoarchitecture and organ of Corti degeneration. (**A**) Audiograms representing click and tone burst stimuli (4, 8, 16, 24 and 32 kHz) auditory brainstem response (ABR) thresholds of 8- and 16-week-old wild-type (WT), *Dusp1* knock-out (KO) and NAC-treated *Dusp1* KO mice. Data are presented as mean ± SEM of at least 15 mice per condition. Statistical significance between genotypes was analyzed by one-way ANOVA: * vs. KO, # vs. WT (*, # *p* < 0.05; **, ## *p* < 0.01; *** *p* < 0.001). (**B**) Representative basal, middle and apical turns hematoxylin-eosin stained cochlear mid-modiolar microphotographs of the organ of Corti (OC) (a–c, g–i and m–o) and spiral ganglion (SG) (d–f, j–l and p–r). SG close-ups (d’–f’, j’–l’ and p’–r’). Scale bars: 25 μm in a, 50 μm in d and 25 μm in d’. Asterisks and arrowheads indicate the absence or presence, respectively, of hair cells and fibers. (**C**) Representative confocal images of TUNEL-stained (green) basal, middle and apical basilar membrane regions of 16-week-old WT, *Dusp1* KO and NAC-treated *Dusp1* KO mice co-immunolabeled with Myo7a (red). Asterisks and arrowheads indicate hair cell (HC) loss and TUNEL^+^ nuclei, respectively. Scale bar: 50 μm. (**D**) Inner (IHC) and outer (OHC) hair cell number per 200 μm of basal, middle or apical basilar membrane regions. (**E**) Percentage of TUNEL^+^ OHCs in 200 μm of basal, middle or apical basilar membrane sections. Data are presented as mean ± SEM of 3 mice per condition. Statistical significance between genotypes was analyzed by one-way ANOVA: * vs. KO, # vs. WT (*, # *p* < 0.05; **, ## *p* < 0.01; *** *p* < 0.001).

**Figure 4 antioxidants-10-01351-f004:**
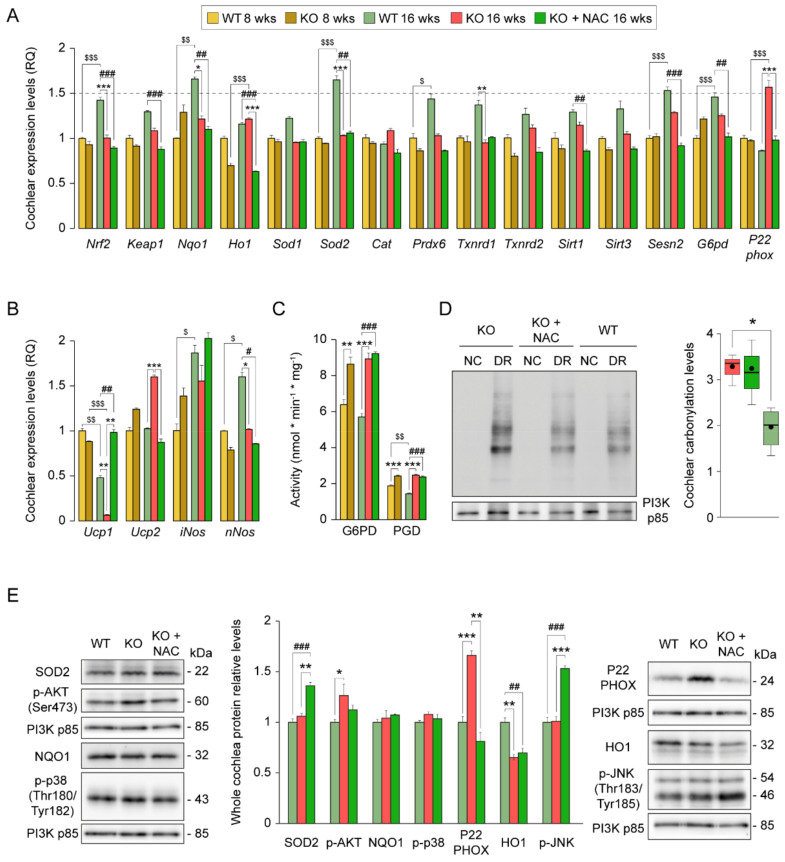
Impact of N-acetylcysteine administration for cochlear redox homeostasis. (**A**,**B**) RT-qPCR gene expression levels of redox enzymes from whole cochlea pooled samples from 3 mice per condition. Expression levels were calculated as 2^−ΔΔCt^ (RQ) from triplicate measurements using *Hprt1* as a reference gene and normalized to levels in 8-week-old wild-type (WT) mice. Data are presented as mean ± SEM. (**C**) Glucose-6-phosphate dehydrogenase (G6PD) and 6-phosphogluconate dehydrogenase (PGD) activity from pooled inner ear cytosolic fractions of two mice per condition. Measurements were performed in triplicate, and values are presented relative to milligram of protein. Data are presented as mean ± SEM. (**D**) Cochlear oxidative protein carbonylation levels of pooled cochlear protein extracts from 3 mice per condition analyzed with the Oxyblot™ Kit. Derivatized protein extracts (DR) are shown together with non-derivatized extracts (NC) in the representative blot image. Quantification was performed in triplicate, and levels were referred to those of PI3K. Data are presented as a box plot, mean value is plotted as a filled circle and whiskers represent min and max values. (**E**) Western blotting of pooled cochlear protein extracts from 3 mice per condition. All protein levels are referred to those of PI3K and normalized to data from 16-week-old WT mice. Data are presented as mean ± SEM. Statistical significance between genotypes and time points was analyzed by one-way ANOVA: * vs. KO, # vs. WT, $ 8-week-old mice vs. 16-week-old mice (*, #, $ *p* < 0.05; **, ##, $$ *p* < 0.01; ***, ###, $$$ *p* < 0.001).

**Figure 5 antioxidants-10-01351-f005:**
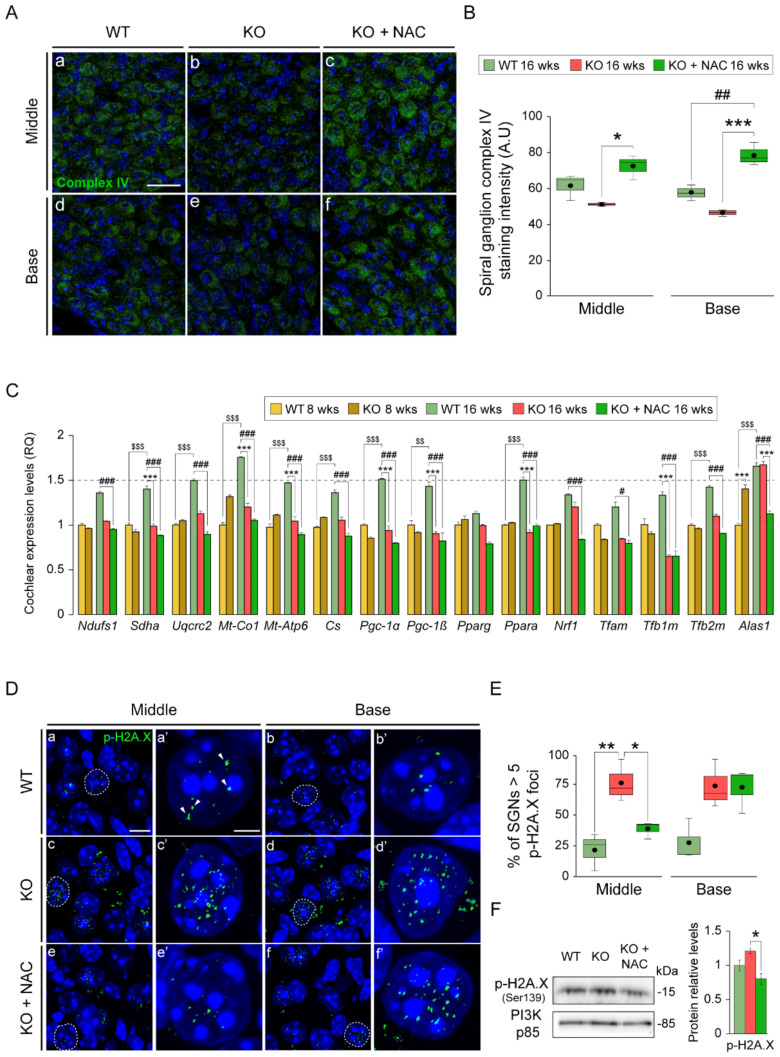
N-acetylcysteine administration preserves mitochondria and reduces DNA damage foci in *Dusp1*^−/−^ spiral ganglion neurons. (**A**) Representative basal and middle turn cochlear spiral ganglion (SG) cross-cryosections of 16-week-old wild-type (WT), *Dusp1* knock-out (KO) and NAC-treated *Dusp1* KO mice immunolabeled for mitochondrial complex IV (green). Scale bar: 20 μm in a. (**B**) Positive stained area intensity of the SG was quantified in 3 mice per condition in at least 3 serial sections per turn. Data are presented as a box plot, mean value is plotted as a filled circle and whiskers represent min and max values. (**C**) RT-qPCR gene expression levels of mitochondrial proteins and biogenesis program genes from whole cochlea pooled samples from 3 mice per condition. Expression levels were calculated as 2^−ΔΔCt^ (RQ) from triplicate measurements using *Hprt1* as a reference gene and normalized to levels in 8-week-old WT mice. Data are presented as mean ± SEM. (**D**) Representative basal and middle turn cochlear SG cross-cryosections of 16-week-old WT, *Dusp1* KO and NAC-treated *Dusp1* KO mice immunolabeled for p-H2A.X in green (a–f). Dashed lines: neuron in close ups (a’–f’). Arrowheads indicate positive foci. Scale bar: 10 μm in a and 2.5 μm in a’. (**E**) Percentage of SG neuron (SGN) nuclei with more than 5 p-H2A.X foci. p-H2A.X foci were counted in 5–10 neuronal nuclei per section in 4 serial sections per turn from at least 3 mice per condition. Data are presented as a box plot, mean value is plotted as a filled circle and whiskers represent min and max values. (**F**) Western blotting of pooled cochlear protein extracts from 3 mice per condition. p-H2A.X data are relative to those for PI3K and normalized to data from 16-week-old WT mice. Data are presented as mean ± SEM. Statistical significance between genotypes and time points was analyzed by one-way ANOVA: * vs. KO, # vs. WT, $ 8-week-old mice vs. 16-week-old mice (*, # *p* < 0.05; **, ##, $$ *p* < 0.01; ***, ###, $$$ *p* < 0.001).

**Figure 6 antioxidants-10-01351-f006:**
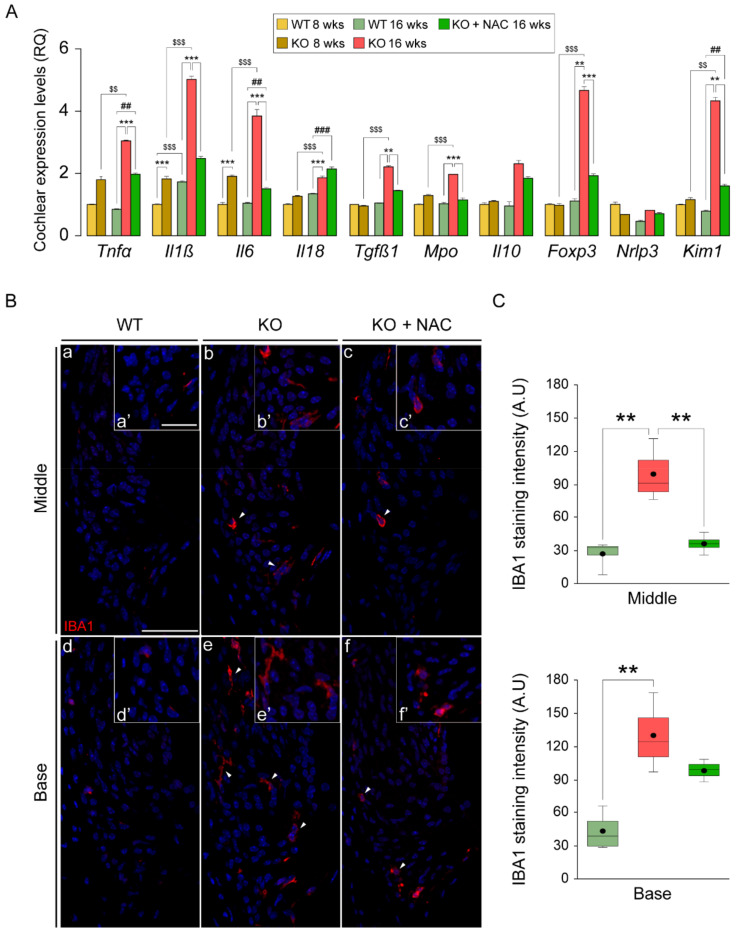
Cochlear exacerbated inflammatory response and macrophage recruitment in *Dusp1*^−/−^ mice is limited by N-acetylcysteine administration. (**A**) RT-qPCR gene expression levels of inflammatory mediators from whole cochlea pooled samples from 3 mice per condition. Expression levels were calculated as 2^−ΔΔCt^ (RQ) from triplicate measurements using *Hprt1* as a reference gene and normalized to levels in 8-week-old wild-type (WT) mice. Data are presented as mean ± SEM. (**B**) Representative basal and middle turn cochlear spiral ligament (Spl) cross-cryosections of 16-week-old WT, *Dusp1* knock-out (KO) and NAC-treated *Dusp1* KO mice immunolabeled for IBA1 in red (a–f). Arrowheads highlight positive macrophages staining. Spl close ups (a’–f’). Scale bar: 40 μm in a and 25 a’. (**C**) IBA1 quantification in the spiral ligament. Staining intensity was quantified in 3 mice per condition in at least 3 serial sections per turn. Data are presented as a box plot, mean value is plotted as a filled circle and whiskers represent min and max values. Statistical significance between genotypes and time points was analyzed by one-way ANOVA: * vs. KO, # vs. WT, $ 8-week-old mice vs. 16-week-old mice (**, ##, $$ *p* < 0.01; ***, ###, $$$ *p* < 0.001).

## Data Availability

RNAseq data discussed in this publication have been deposited in NCBI’s Gene Expression Omnibus [70] and are accessible through GEO Series accession number GSE176114 (https://www.ncbi.nlm.nih.gov/geo/query/acc.cgi?acc=GSE176114). Additional data that support the findings of this study are contained within the article.

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
