# Peer review of "Dual-Specificity Phosphatase 1 (DUSP1) Has a Central Role in Redox Homeostasis and Inflammation in the Mouse Cochlea"

_antioxidants, 2021, doi:10.3390/antiox10091351_

Round 1

Reviewer 1 Report

The manuscript is written in a detailed and understandable way. The research was carried out at a high level, a variety of methods are used and the conclusions are consistent with the results obtained. The study is not fundamentally new, since the antioxidant properties of N-acetylcystein have already been shown. If NAC was used as a positive control, authors need to describe this more clearly in the Introduction. Also I have a small comment on the text formatting. The section "materials and methods" should be written in accordance with the Antioxidants style.

Reviewer 2 Report

The manuscript is well organized, results are clearly presented and references are up-to-date. There are some minor issues that should be addressed by authors.

Page 2, Lines 49, 52, 59, 74 and throughout the manuscript: Please make clear distinction between in vivo (animal and human studies) and in vitro studies.

Page 2, Line 52: Use of antioxidants is oral? What about the dose?

Page 5, Line 241: Please convert rpm to g.

Reviewer 3 Report

The findings are interesting and technically well performed. Specific points that the authors need to address are as follows:

  1. How N-acetylcysteine treatment delayed the onset of SNHL and mitigated cochlear damage should be investigated in detail. It will be interesting to investigate if similar effects can be obtained with other antioxidants.
  2. How N-acetylcysteine inhibited cytokine production and reduced macrophage recruitment should be analysed.
  3. It is not clear that why NADPH production was not significantly reduced in NAC-treated cochlea. This finding appears to be contradictory and should be re-examined.
  4. The authors should provide their own justification and relevance of the study. This will help the readers to understand the importance of the paper.
  5. Minor typographical errors were found throughout the manuscript and should be corrected.

Round 2

Reviewer 1 Report

The authors answered my question convincingly in the cover letter, but the manuscript was not revised enough. It is necessary to cite this text from cover letter in the Introduction section:

 "Our work aim was to deepen in the understanding of the role of DUSP1 in hearing loss, which we began to unveil in our previous paper (Celaya, et al., 2019). The comparative RNAseq analysis of the cochleae of wild type and mutant Dusp1 knock out mice shown in this manuscript, suggested that oxidation secondary to DUSP1 loss of action had a role in triggering premature hearing loss. The working hypothesis was that if the redox imbalance was significant, treatment with an antioxidant should prevent hearing loss. N-acetylcysteine (NAC) was used because its antioxidant properties are well-known and its human use approved. Therefore, NAC was used as a pharmacological tool to prevent redox imbalance."

This paragraph is really needed in the Introduction because the authors in the Introduction did not clearly formulate the goals and objectives of the manuscript, but indicated the results already obtained.

Round 3

Reviewer 1 Report

I have no more comments.

Author Response

We are grateful with reviewer peer review that have improved the quality and clarity of the manuscript.